

# Morphometric and microsatellite-based comparative genetic diversity analysis in *Bubalus bubalis* from North India

Vikas Vohra[1], Narendra Pratap Singh[1], Supriya Chhotaray[1],
Varinder Singh Raina[2], Alka Chopra[3] and Ranjit Singh Kataria[4]

[1] Animal Genetics and Breeding Division, National Dairy Research Institute, Karnal, Haryana, India
[2] Department of Animal Husbandry and Dairying, Ministry of Fisheries, Animal Husbandry and Dairying, New Delhi, New Delhi, India
[3] Animal Biotechnology Centre, National Dairy Research Institute, Karnal, Haryana, India
[4] Animal Biotechnology Division, ICAR—National Bureau of Animal Genetic Resources, Karnal, Haryana, India

Corresponding author
Vikas Vohra,
Vikas.Vohra@icar.gov.in

## ABSTRACT

To understand the similarities and dissimilarities of a breed structure among different buffalo breeds of North India, it is essential to capture their morphometric variation, genetic diversity, and effective population size. In the present study, diversity among three important breeds, namely, Murrah, Nili-Ravi and Gojri were studied using a parallel approach of morphometric characterization and molecular diversity. Morphology was characterized using 13 biometric traits, and molecular diversity through a panel of 22 microsatellite DNA markers recommended by FAO, Advisory Group on Animal Genetic Diversity, for diversity studies in buffaloes. Canonical discriminate analysis of biometric traits revealed different clusters suggesting distinct genetic entities among the three studied populations. Analysis of molecular variance revealed 81.8% of genetic variance was found within breeds, while 18.2% of the genetic variation was found between breeds. Effective population sizes estimated based on linkage disequilibrium were 142, 75 and 556 in Gojri, Nili-Ravi and Murrah populations, respectively, indicated the presence of sufficient genetic variation and absence of intense selection among three breeds. The Bayesian approach of STRUCTURE analysis (at $K = 3$) assigned all populations into three clusters with a degree of genetic admixture in the Murrah and Nili-Ravi buffalo populations. Admixture analysis reveals introgression among Murrah and Nili-Ravi breeds while identified the Gojri as unique buffalo germplasm, indicating that there might be a common origin of Murrah and Nili-Ravi buffaloes. The study provides important insights on buffalo breeds of North India that could be utilized in designing an effective breeding strategy, with an appropriate choice of breeds for upgrading local non-descript buffaloes along with conservation of unique germplasm.

## INTRODUCTION

Through the course of evolution, forces such as mutation, adaptation, reproductive isolation, random drift, selection and breeding have created vast diversity (breeds) among the *Bubalus bubalis* in India. Many well-defined breeds have been formed for various purposes and the high-performance breeds are intensely selected worldwide. This has led to the replacement of native low performance breeds with high performance ones causing erosion of genetic resources (*Groeneveld et al., 2010*). However, future trait of interests concerns the genetic diversity of low-performance breeds so, that the population can be maintained for future breeding (*Notter, 1999*). The genetic diversity of livestock species such as cattle, buffalo, pig, sheep, goat, camel and chicken have been widely studied in several population and the diversity of zebu and taurine cattle breeds is one of the most studied. Diversity between swamp and riverine buffaloes also have been studied using microsatellite markers, mitochondrial D-loop and cytochrome b sequence variations (*Barker et al., 1997*; *Lau et al., 1998*; *Zhang et al., 2007*).

Over the past few decades, world buffalo population has rapidly increased with 208 million buffaloes in the world presently, having 96.79% of population in Asia, 1.68% in Africa, 1.23% in the Americas and 0.22% in Europe (*FAOSTAT, 2019*). India alone has 109.85 million of buffalo population with 17 registered breeds (*Livestock Census, 2019*). Water buffalo (*Bubalus Bubalis*) that most probably domesticated in Indus Valley region for multiple utility creates a rich Bubaline diversity in Northern regions of India, comprising states of Punjab, Haryana, Himachal Pradesh, Delhi and Western Uttar Pradesh. The predominant bubaline genetic resources documented from the region include Murrah, Nili Ravi and Gojri buffaloes (http://www.nbagr.res.in/regbuf.html). Murrah being dominating buffalo germplasm with superior milk-producing ability has suppressed the need for identification and characterization of other breeds. On the other hand, Gojri is one of the little-known buffalo population of the region, with a good milch potential on low to zero input system of dairying and is maintained on a semi-migratory extensive system of management (*Vohra, Niranjan & Joshi, 2012*; *Vohra et al., 2015*).

Characterization and classification of animal genetic resources (AnGR) require ample knowledge of the geographical distribution of the breeds, identification of unique characteristics, population size and structure, production environment, and genetic diversity. It is customary to perform a detailed molecular study along with physical and phenotypic assessment to check within and between population diversity in order to characterize a population (*Weitzman, 1993*; *Hall & Bradley, 1995*; *Barker, 1999*; *Ruane, 2000*; *Bruford, Bradley & Luikart, 2003*; *Simianer, 2005*; *Toro & Caballero, 2005*). *Vohra et al. (2015)* have used 13 morphometric traits of Gojri buffaloes for phenotypic characterization using Principal Component Analysis, a multivariate statistical technique. Multivariate statistical analysis techniques viz. classical principal component analysis serves the objectives of dimension reduction and clustering when multiple morphometric traits are measured (*Johnson & Wichern, 2002*; *Yadav, Arora & Jain, 2017*).

The neutrality, co-dominant inheritance and high polymorphic information content of microsatellite markers have rendered them as the markers of choice for diversity studies

(*Metta et al., 2004*; *Li et al., 2005*; *Yoon et al., 2005*; *Sodhi et al., 2005*; *Kumar et al., 2006*; *Pandey et al., 2006a*, *2006b*; *Vijh et al., 2008*; *Sharma et al., 2013*). Several genetic diversity studies of water buffalo populations have been carried out throughout the world using microsatellite markers (*Kataria et al., 2009*; *Ángel-Marín et al., 2010*; *Gargani et al., 2010*; *Mekkawy et al., 2012*; *Özkan Ünal et al., 2014*; *Vohra et al., 2017*; *Khade et al., 2019*).

The genetic diversity within Murrah, Nili-Ravi and Gojri breeds have been studied independently that share the common breeding tract in North India. However, in India, buffalo breeding is largely restricted to natural mating that subsequently may have led to admixture of these populations. Hence, there is a need to assess the between-breed genetic diversity among these breeds. The present study was performed to assess the levels of genetic diversity, and population structure among three buffalo breeds of North India. The results will help in formulating an effective breeding, management policy, shaping future conservation plans for maintaining breed purity and reducing the possible admixture due to introgression among purebreds. Thus, it is imperative to compare the region-specific diversity and breed status of bubaline germplasm.

## MATERIALS & METHODS

### Sampling strategy

Sampling was done from their respective native tracts, to compare the genetic diversity between three different breeds. Gojri buffalo samples were collected during 2017–18 from areas of Punjab and Himachal Pradesh (30° 9′ to 32° 3′ N and 75° to 77° E) states of India, and samples for Nili-Ravi buffaloes were collected from Punjab state ( 28° 17′ to 32° 17′ N and 74° to 76° 41′ E). The Nili-Ravi has a comparatively smaller geographical distribution compared to Murrah and Gojri. In India, Murrah buffaloes are found in almost all regions but its native area is Haryana state (28° 02′ to 30° 21′ N and 75° to 77° E) hence, sampling was performed from Haryana and Punjab. The data of Murrah and Nili-Ravi was taken for comparative analysis from Buffalo Genomics Lab of National Bureau of Animal Genetic Resources, Karnal. The breeding and sampling tract had a herd size of 2–6 buffaloes per households. To ensure that selected animals are unrelated, in the absence of detailed pedigree accounts, buffalo breeders were interviewed in detail and their records were checked. Only those animals who were not having common parents for at least 3–4 generations were included in the study. Buffaloes were selected for this study following guidelines of measurement of domestic animal diversity program (*FAO, 2011*) those represented the original indigenous true to type phenotype. Blood samples were collected with the consent of herd owners. Approximately 5–10 ml of blood from jugular vein was collected by trained Veterinarian using aseptic measures. All the studies were carried out under approval of ICAR-National Dairy Research Institute IAEC 1705/GO/ac/13/CPCSEA.

Morphometric traits were measured on a total of 242 adult female buffaloes, comprising of 113 Murrah, 37 Nili-Ravi and 92 Gojri buffaloes, to avoid the sex and age differences. Thirteen (13) different traits were measured on all three breeds *De Melo et al. (2018)*. All the measurements on the animal were recorded in their normal standing position on a levelled surface using a tape measure by the same technical person. Traits

recorded were body height (HT), body length (BL), chest girth (CG), paunch girth (PG), face length (FL), face width (FW), horn length (HL), horn circumference (HC), ear length (EL), distance between hip bone (HB), distance between pin bone (PB), tail length (TL), and tail length up to switch (TS). To avoid age effects, only adult buffaloes (3.5 years above) were included in study. For microsatellite genotyping, blood samples were collected from 128 (40 Murrah, 40 Nili-Ravi and 48 Gojri) buffaloes.

## Genotyping microsatellite markers

Genomic DNA was isolated from blood samples by standard phenol–chloroform extraction protocol, as described by *Sambrook & Russel (2001)*. DNA concentration was checked by spectrophotometric method. Genetic variation was assayed using 25 microsatellite markers. Microsatellite genotyping was carried out as previously describe in *Vohra et al. (2017)* following the protocol of *Mishra et al. (2010)*. Fluorescent-tagged forward primers for each microsatellite were used. The primers those were able to produce a fragment size >75 bp were used in the study. Fragment length analysis was performed through ABI PRISM 3100 automatic sequencer (Applied Biosystems, Foster City, CA, USA) after performing polymerase chain reaction (PCR) for fragment amplification. Allele length for the different fragments generated was determined as described in *Vohra et al. (2017)* using GeneScan software (version 5.0; Applied Bio system, Foster City, CA, USA). Observed number of alleles ($N_a$), theta estimate ($\theta_H$), expected heterozygosity ($H_e$), $F_{IT}$ (total inbreeding estimate), $F_{ST}$ (measurement of population differentiation) and $F_{IS}$ (within- population-inbreeding estimate) were calculated using Arlequin v3.5 (*Excoffier & Lischer, 2010*). Pairwise differences between populations using molecular distances were calculated. Molecular diversity indices were calculated as per *Tajima (1983, 1993)*, *Nei (1987)* and *Zouros (1979)*, implemented in Arlequin v3.5, and allowing 5% level of missing data. Analysis of molecular variances was done using 1,000 permutations. Exact test of population differentiation was performed with 1,00,000 Markov chain steps and 10,000 dememorization steps.

## Statistical analysis

Statistical analyses on morphometric data were performed using SPSS v17.0 software (*SPSS, 2001*). Multivariate analysis technique such as canonical discriminant analysis (CDA) simultaneously analyse multiple correlated measurements in a single individual and increases the discriminatory power by eliminating variables explaining less variation in the dataset. The relation between the group the individual belongs to and a set of morphometric traits are quantified using CDA (*Zhao & Maclean, 2000*). As, CDA provides optimum discrimination between population to classify them as a different breed hence, widely used in breed characterization and genetic diversity studies.

The canonical discriminant analysis was performed in SAS v9.3 program (*SAS Institute Inc., 2011*) using Proc disc procedure, for determining the most discriminatory morphometric traits. The probabilities of assigning an individual to a population were determined using Discrim procedure based on the linear discriminant function that included the thirteen morphometric variables. Wilk's Lambda was used as the test statistics

to check for the differences between the means of identified groups of subjects on a combination of dependent variables.

Population assignment was performed using the Bayesian Markov chain Monte Carlo approach implemented in Structure v2.3.4 (*Pritchard, Stephens & Donnelly, 2000*). The Bayesian clustering algorithm simultaneously estimates allele frequencies at each and individuals are assigned probabilistically to one of the $K$ subpopulations. It assumes that prior distribution of population to which individuals belong and allele frequencies are known.

The most likely number of subpopulations was determined by the Evanno $\Delta K$ method (*Evanno, Regnaut & Goudet, 2005*) using R package "POPHELPER" (*Francis, 2017*). Twenty independent runs were performed for $K = 2$ to 4 to identify the most likely number of clusters present in the dataset. The analysis was performed with a burn in period of 10,000 and 50,000 MCMC iterations. Effective population size ($N_e$) was checked for the three population. $N_e$ was estimated using linkage disequilibrium method using NeEstimator v2.01 (*Do et al., 2014*) Software. The P-critical value (rare allele frequency) was set to 0.05, below which all the alleles were rejected. Jackknife confidence intervals (CI) were calculated for each estimate, $N_e$, of different population. Discrimination between populations was elucidated graphically through principal coordinate analysis (PCoA) using Darwin v6.0.021 (*Perrier, Flori & Bonnot, 2003*). Principal coordinate analysis is a classical multidimensional scaling method based on dissimilarity or distance matrix to assign each individual a location in a two or three-dimensional space. The dissimilarity matrix based phylogenetic tree was also obtained through Darwin.

## RESULTS

### Classificatory analysis based on morphometric traits

The means and standard deviation, coefficient of variations and comparison of mean difference between populations for each trait across population is listed in Table 1. A Canonical Discriminant analysis was used to compare different morphometric traits and first two canonical discriminant functions were used in the analysis, which explained 66.7% and 33.3% of total variance, respectively. Wilk's Lambda was used as the test statistics to check the difference between means of the two groups and was found to be significant (Table 2). Classification based on canonical discriminant functions for both original and cross-validated counts predicted 100% assignment of each adult buffaloes to their hypothetically known populations *i.e.* Murrah, Nili-Ravi and Gojri. All the individuals plotted based on 1st and 2nd canonical discriminant functions were clustered into three distinct groups suggesting three different breeds in the sample (Fig. 1).

### Microsatellite variations

Among 25 microsatellite loci genotyped for this study, only 22 loci that were polymorphic for all three populations were used for further downstream analysis. A total of 145, 138 and 173 alleles were found across 22 loci in the 128 individuals sampled from the Murrah, Nili-Ravi, and Gojri buffaloes, respectively. ILSTS60 was highly polymorphic in Gojri buffaloes, ILSTS95 in both Murrah and Nili-Ravi and ILSTS61 in Murrah (Fig. S1A). Mean

**Table 1 Average measurements of body morphometric traits in 3 buffalo populations from Northern India.**

| Traits (measured in cm) | Pop | Mean ± SE | SD | Min. | Max. | CV% | Pop (i) | Pop (j) | Mean Difference (i-j) | p-Value |
|---|---|---|---|---|---|---|---|---|---|---|
| Body height | Mu | 138.40 ± 0.41 | 4.40 | 129.00 | 150.00 | 3.17 | Mu | NR | 4.29 ± 0.85 | 0.095 |
| | NR | 134.10 ± 1.10 | 4.79 | 108.00 | 134.00 | 4.35 | Mu | Goj | 9.58* ± 0.63 | 0.0001 |
| | Goj | 128.82 ± 0.47 | 4.51 | 118.00 | 145.00 | 3.49 | NR | Goj | −5.28* ± 0.88 | 0.0001 |
| Body length | Mu | 129.26 ± 0.55 | 5.85 | 115.00 | 147.00 | 4.52 | Mu | NR | 23.45* ± 1.08 | 0.0001 |
| | NR | 105.81 ± 1.12 | 6.83 | 91.00 | 121.00 | 6.45 | Mu | Goj | −4.22* ± 0.80 | 0.0001 |
| | Goj | 133.48 ± 0.51 | 4.90 | 122.00 | 151.00 | 3.67 | NR | Goj | −27.67* ± 1.11 | 0.0001 |
| Chest girth | Mu | 212.53 ± 1.12 | 11.87 | 185.00 | 250.00 | 5.58 | Mu | NR | 51.37* ± 2.01 | 0.0001 |
| | NR | 161.16 ± 1.52 | 9.27 | 144.00 | 182.00 | 5.74 | Mu | Goj | 16.63* ± 1.49 | 0.0001 |
| | Goj | 195.90 ± 0.98 | 9.43 | 170.00 | 214.00 | 4.81 | NR | Goj | −34.74* ± 2.07 | 0.0001 |
| Paunch girth | Mu | 232.11 ± 1.09 | 11.54 | 208.00 | 266.00 | 4.97 | Mu | NR | 60.84* ± 2.84 | 0.0001 |
| | NR | 171.27 ± 1.79 | 10.89 | 153.00 | 198.00 | 6.35 | Mu | Goj | 18.91* ± 2.11 | 0.0001 |
| | Goj | 213.20 ± 2.03 | 19.50 | 121.00 | 242.00 | 9.14 | NR | Goj | −41.92* ± 2.92 | 0.0001 |
| Face length | Mu | 49.29 ± 0.25 | 2.61 | 46.00 | 62.00 | 5.27 | Mu | NR | 8.40* ± 0.41 | 0.0001 |
| | NR | 40.89 ± 0.31 | 1.88 | 38.00 | 46.00 | 4.60 | Mu | Goj | 0.66 ± 0.31 | 0.081 |
| | Goj | 48.63 ± 0.17 | 1.66 | 44.00 | 54.00 | 3.41 | NR | Goj | −7.74* ± 0.43 | 0.0001 |
| Face width | Mu | 19.72 ± 0.09 | 0.98 | 18.00 | 22.00 | 4.92 | Mu | NR | −0.15 ± 0.51 | 0.955 |
| | NR | 19.86 ± 0.19 | 1.18 | 18.00 | 23.00 | 5.95 | Mu | Goj | −3.29* ± 0.38 | 0.0001 |
| | Goj | 23.01 ± 0.43 | 4.17 | 20.00 | 49.00 | 18.10 | NR | Goj | −3.15* ± 0.53 | 0.0001 |
| Ear length | Mu | 28.41 ± 0.11 | 1.22 | 25.00 | 30.00 | 4.29 | Mu | NR | 7.92* ± 0.23 | 0.0001 |
| | NR | 20.49 ± 0.17 | 1.04 | 19.00 | 22.00 | 5.09 | Mu | Goj | −0.30 ± 0.17 | 0.110 |
| | Goj | 28.75 ± 0.13 | 1.25 | 21.00 | 31.00 | 4.34 | NR | Goj | −8.26* ± 0.24 | 0.0001 |
| Horn Length | Mu | 28.37 ± 0.34 | 3.59 | 16.00 | 34.00 | 12.61 | Mu | NR | −17.84* ± 1.19 | 0.0001 |
| | NR | 46.22 ± 0.88 | 5.32 | 34.00 | 56.00 | 11.51 | Mu | Goj | −16.36* ± 0.88 | 0.0001 |
| | Goj | 44.73 ± 0.91 | 8.73 | 23.00 | 82.00 | 19.52 | NR | Goj | 1.49 ± 1.22 | 0.443 |
| Horn circumference | Mu | 17.18 ± 0.16 | 1.70 | 12.00 | 21.00 | 9.90 | Mu | NR | −2.31* ± 0.33 | 0.0001 |
| | NR | 19.49 ± 0.29 | 1.76 | 17.00 | 25.00 | 9.02 | Mu | Goj | −2.69* ± 0.24 | 0.0001 |
| | Goj | 19.87 ± 0.18 | 1.73 | 17.00 | 28.00 | 8.70 | NR | Goj | −0.38 ± 0.34 | 0.488 |
| Hip bone | Mu | 55.55 ± 0.29 | 3.13 | 49.00 | 63.00 | 5.61 | Mu | NR | 15.95* ± 0.6 | 0.0001 |
| | NR | 39.59 ± 0.36 | 2.20 | 34.00 | 43.00 | 5.56 | Mu | Goj | 1.95* ± 0.45 | 0.0001 |
| | Goj | 53.60 ± 0.37 | 3.53 | 30.00 | 60.00 | 6.69 | NR | Goj | −14.00* ± 0.62 | 0.0001 |
| Pin bone | Mu | 16.94 ± 0.10 | 1.08 | 15.00 | 20.00 | 6.32 | Mu | NR | 2.83* ± 0.54 | 0.0001 |
| | NR | 14.11 ± 0.30 | 1.81 | 10.00 | 17.00 | 12.80 | Mu | Goj | −7.37* ± 0.40 | 0.0001 |
| | Goj | 24.30 ± 0.45 | 4.31 | 19.00 | 59.00 | 17.69 | NR | Goj | −10.20* ± 0.55 | 0.0001 |
| Tail length | Mu | 104.61 ± 1.24 | 13.18 | 76.00 | 130.00 | 12.58 | Mu | NR | 24.26* ± 2.59 | 0.0001 |
| | NR | 80.35 ± 1.06 | 6.42 | 69.00 | 102.00 | 7.99 | Mu | Goj | 13.65* ± 1.92 | 0.0001 |
| | Goj | 90.96 ± 1.68 | 16.11 | 22.00 | 116.00 | 17.70 | NR | Goj | −10.60* ± 2.66 | 0.0001 |
| Tail up to switch | Mu | 95.14 ± 1.04 | 11.01 | 68.00 | 123.00 | 11.56 | Mu | NR | 25.87* ± 1.84 | 0.0001 |
| | NR | 69.27 ± 0.98 | 5.93 | 60.00 | 85.00 | 8.56 | Mu | Goj | −9.14* ± 1.37 | 0.0001 |
| | Goj | 104.28 ± 0.97 | 9.27 | 73.00 | 124.00 | 8.87 | NR | Goj | −35.01* ± 1.9 | 0.0001 |

**Notes:**
* The mean difference is significant at the 0.05 level.
Pop, Mu, NR, Goj in the table corresponds to 'Population', 'Murrah', 'Nili-Ravi' and 'Gojri', respectively.

**Table 2 Characteristics of canonical discriminant functions and test statistics.**

| Discriminant function | Eigenvalues | Variance percentage explained | Cumulative variance | Canonical correlation | Wilks' Lambda | Chi-square | †d.f. | P |
|---|---|---|---|---|---|---|---|---|
| 1st function | 14.40 | 66.7 | 66.7 | 0.967 | 0.008 | 1134.64 | 20 | 0.0001 |
| 2nd function | 7.19 | 33.3 | 100 | 0.937 | 0.122 | 493.30 | 9 | 0.0001 |

Note:
† Degrees of freedom.

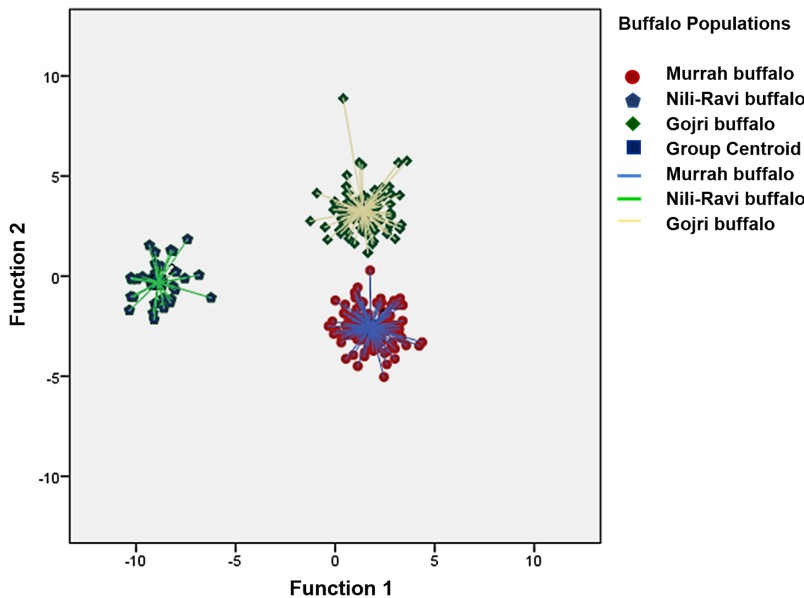

**Figure 1 Canonical Discriminant Analysis (Scatter Plot) based on 13 body morphometric traits depicted three different buffalo populations from Northern India.**

number of alleles for all populations varied from 3.67 ± 2.08 at ILSTS19 to 10.33 ± 0.58 at CSSM47. Mean expected heterozygosity ($H_e$) across all populations ranged from 0.14 ± 0.02 at ILSTS19 to 0.81 ± 0.04 at ILSTS58. The mean $H_e$ estimated over all loci was lowest in Murrah (0.58 ± 0.25) while it was highest in Gojri population (0.70 ± 0.15) (Fig. S1B). Estimator of mutation parameter ($\theta_H$) that is obtained using observed homozygosity values was estimated under infinite allele model. Mean $\theta_H$ ranged from 1.36 in Murrah to 2.33 in Gojri buffaloes (Fig. S1C). Across all three populations mean $\theta_H$ ranged from 0.17 ± 0.03 (ILSTS19) to 4.44 ± 1.16 (ILSTS58). Marker wise number of alleles, $H_e$, and $\theta_H$ in each breed given in Table 3.

## Genetic diversity

Global Analysis of molecular variance (AMOVA) using 19 polymorphic loci was accomplished. Wright's F-statistics values obtained from the results of global AMOVA revealed 11.7% deficit of heterozygotes for each of the analyzed breeds ($F_{IS}$) whereas the total population had a 27.8% deficit of heterozygotes ($F_{IT}$). The average genetic differentiation ($F_{ST}$) between the breeds was 18.2% ($p = 0.00001$) indicating significantly

**Table 3 Breed wise details of estimated Genetic Diversity Indices for each microsatellite markers.**

| Locus | Expected Heterozygosity ($H_e$) | | | Theta H ($\theta_H$) | | | Number of alleles ($N_a$) | | | $F_{ST}$ | p-Value of $F_{ST}$ estimation |
|---|---|---|---|---|---|---|---|---|---|---|---|
| | Nilli-Ravi | Murrah | Gojri | Nilli-Ravi | Murrah | Gojri | Nilli-Ravi | Murrah | Gojri | | |
| BM1818 | 0.69 | 0.68 | 0.71 | 2.29 | 2.17 | 2.49 | 7 | 9 | 6 | 0.029 | 0.078 |
| CSSM19 | 0.74 | 0 | 0.73 | 2.80 | 0 | 2.75 | 6 | 0 | 6 | NE | NE |
| CSSM33 | 0.71 | 0.73 | 0.63 | 2.49 | 2.67 | 1.74 | 8 | 9 | 7 | 0.194 | 0.00001 |
| CSSM45 | 0.65 | 0.80 | 0.73 | 1.87 | 3.98 | 2.72 | 5 | 6 | 6 | 0.204 | 0.00001 |
| CSSM47 | 0.81 | 0.69 | 0.82 | 4.23 | 2.25 | 4.63 | 10 | 10 | 11 | 0.175 | 0.00001 |
| CSSM66 | 0.79 | 0.61 | 0.82 | 3.67 | 1.60 | 4.43 | 7 | 6 | 9 | 0.176 | 0.00001 |
| Hel013 | 0.67 | 0.75 | 0.82 | 2.08 | 3 | 4.49 | 8 | 8 | 9 | 0.139 | 0.00001 |
| ILSTS19 | 0.14 | 0.17 | 0.12 | 0.16 | 0.20 | 0.14 | 2 | 6 | 3 | 0.809 | 0.00001 |
| ILSTS25 | 0.58 | 0.61 | 0.72 | 1.39 | 1.59 | 2.52 | 5 | 6 | 8 | 0.179 | 0.00001 |
| ILSTS26 | 0.67 | 0.61 | 0.76 | 2.05 | 1.57 | 3.26 | 6 | 6 | 5 | 0.084 | 0.00001 |
| ILSTS28 | 0.76 | 0.76 | 0.76 | 3.25 | 3.16 | 3.21 | 8 | 7 | 6 | 0.002 | 0.7165 |
| ILSTS29 | 0.33 | 0.26 | 0.82 | 0.49 | 0.35 | 4.49 | 6 | 4 | 10 | 0.376 | 0.00001 |
| ILSTS30 | 0.71 | 0.60 | 0.69 | 2.45 | 1.47 | 2.25 | 7 | 6 | 6 | 0.194 | 0.00001 |
| ILSTS33 | 0.66 | 0.68 | 0.59 | 1.99 | 2.14 | 1.47 | 6 | 7 | 3 | 0.005 | 0.51026 |
| ILSTS36 | 0.66 | 0.67 | 0.72 | 1.98 | 2.03 | 2.58 | 6 | 6 | 12 | 0.160 | 0.00001 |
| ILSTS52 | 0.67 | 0 | 0.73 | 2.00 | 0 | 2.72 | 9 | 0 | 8 | NE | NE |
| ILSTS56 | 0.40 | 0.53 | 0.59 | 0.67 | 1.12 | 1.45 | 5 | 7 | 8 | 0.345 | 0.00001 |
| ILSTS58 | 0.81 | 0.85 | 0.77 | 4.24 | 5.70 | 3.40 | 7 | 9 | 11 | 0.088 | 0.00001 |
| ILSTS60 | 0.45 | 0.36 | 0.60 | 0.81 | 0.57 | 1.53 | 4 | 3 | 14 | 0.016 | 0.18573 |
| ILSTS61 | 0.66 | 0.81 | 0.76 | 1.91 | 4.23 | 3.23 | 6 | 13 | 11 | 0.178 | 0.00001 |
| ILSTS089 | 0 | 0.79 | 0.76 | 0 | 3.76 | 3.14 | 0 | 6 | 6 | NE | NE |
| ILSTS95 | 0.80 | 0.71 | 0.71 | 4.08 | 2.43 | 2.48 | 10 | 11 | 8 | 0.061 | 0.00001 |
| Mean | 0.61 | 0.58 | 0.70 | 1.55 | 1.36 | 2.33 | 6.27 | 6.59 | 7.86 | 0.18 | – |
| SD | 0.22 | 0.25 | 0.15 | – | – | – | 2.33 | 3.10 | 2.83 | – | – |

**Note:**
NE, not estimable in the global analysis of molecular variance.

higher discrimination between breeds (Table 4). Details of AMOVA results are presented in Table 4. The pair-wise $F_{ST}$, Slatkin linearized $F_{ST}$, and Nei's distance (d) values were used to illustrate the genetic distance between breeds (Fig. 2A, 2B and 2C), which significantly differentiated all three breeds.

Murrah and Nili-Ravi population were clustered together while the Gojri population was present as a distinct group, suggesting it as a different breed in factorial correspondence analysis (Fig. 3) and phylogenetic tree (Fig. S2).

Effective population size ($N_e$) was estimated excluding rare alleles with an allele frequency below 0.05. The estimated effective population size of Gojri, Nili-Ravi and, Murrah was found to be 142, 75 and 556, respectively. The Jack-knife CIs for the $N_e$ estimates were 83–396, 48–141 and 136 to infinity for Gojri, Nili-Ravi and Murrah, respectively at 0.05 P-critical value of rare alleles.

**Table 4 Results of Global Molecular Analysis of Variance (AMOVA) along with fixation indices in Northern India buffalo populations.**

| Sources of variation | Degrees of freedom | Sum of squares | Variance components | Variation explained (%) | Fixation indices | *P* Value |
|---|---|---|---|---|---|---|
| Among populations | 2 | 232.21 | 1.38 | 18.25 | 0.182 ($F_{ST}$) | 0.00001 |
| Among individuals within populations | 125 | 786.65 | 0.73 | 9.60 | 0.117 ($F_{IS}$) | 0.00001 |
| Within individuals | 128 | 640.50 | 5.46 | 72.15 | 0.278 ($F_{IT}$) | 0.00001 |
| Total | 255 | 1659.37 | 7.57 | 100 | – | – |

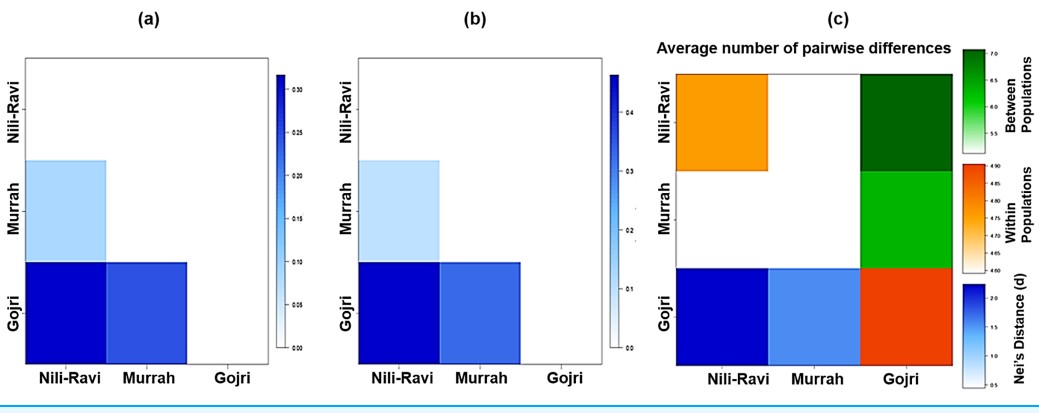

**Figure 2 Heatmap of molecular diversity indices.** (A) Pairwise $F_{ST}$, (B) Slatkin's linearized $F_{ST}$, (C) Nei's distance and AMOVA.

## Bayesian genetic structure

Number of possible sub-populations estimated through Evanno Δ*K* method suggested a maximum of three populations (Fig. 4). Population assignment accomplished in STRUCTURE for *K* = 2, 3 and 4 and results are presented in the form of bar plot (Fig. 5). For *K* = 3, as estimated through Evanno Δ*K* method, it showed 99.4% of Gojri buffaloes are classified into their pre-defined breed. 95.9% of Nili-Ravi and 83.6% of Murrah were assigned to their respective pre-defined groups. Inferred ancestry of each individual (for *K* = 3) along with average proportion of each individuals classified into respective pre-assigned breeds (for *K* = 2, 3 and 4) is reported (Table 5).

## DISCUSSION

In India, limited work on complete characterization and classification of buffalo genetic resources have been carried out in past, primarily due to availability of much acclaimed Murrah buffaloes. The native breeding tract of Murrah buffalo is North India, and currently, more than 40% of the countries buffalo population is either Murrah or has been crossed with Murrah buffaloes. Hence, genetic studies on other buffalo populations is often neglected. However, several studies have been taken up on morphometric characterization of individual breeds yet there are limited reports on genetic diversity

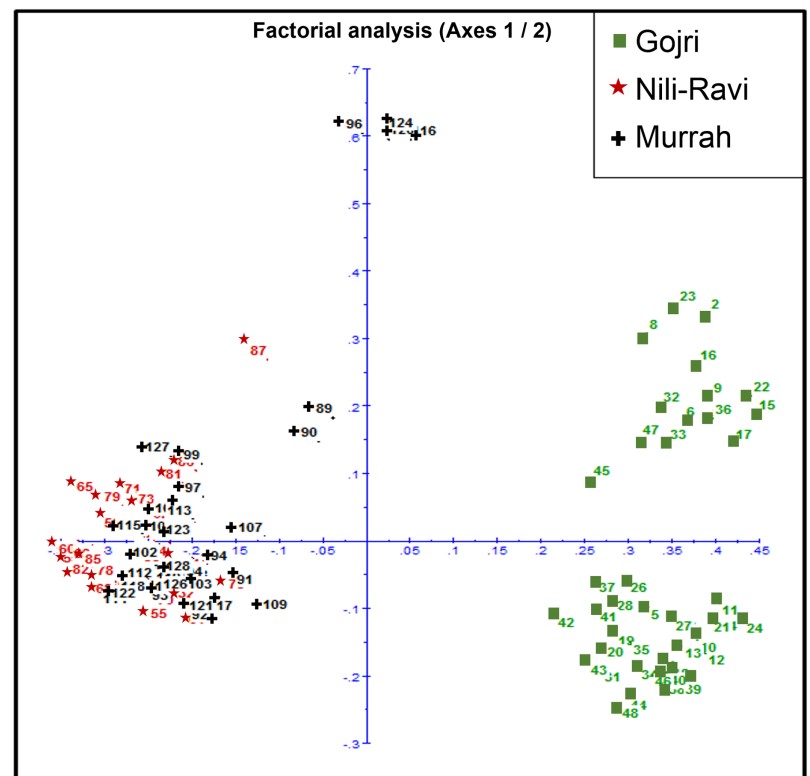

**Figure 3 Scatter plot for Factorial Correspondence Analysis based on genetic diversity indices depicted three different buffalo populations from Northern India.**

studies through molecular markers and comparative studies. A parallel approach of characterization and classification of buffalo germplasm in a region is much needed for genetic improvement in such populations. The present study is an evaluation of Riverine buffaloes of North India taking a geographical region-based approach.

## Morphological diversity

Gojri animals with unique phenotypic appearance are quite distinct from Murrah, Murrah crosses, and Nili Ravi (*Vohra, Niranjan & Joshi, 2012*). The average measurements for body biometric traits across the studied buffalo populations of the North India is listed in Table 1. Thirteen body biometric traits across three population when compared, revealed significant differences among the studied populations, except for FL and EL among Murrah and Gojri buffaloes, HC and HL between Nili-Ravi and Gojri population. Body height (HT) and face width (FW) did not vary among Murrah and Nili-Ravi populations. The comparison of morphometric traits between all three buffalo breeds of the North India outlined the phenotypic distinctness for majority of the body biometric trait. The coefficient of variation (CV) percentage was least for body height in all three breeds. On comparing average of HT, BL, CG and PG, Gojri buffaloes were found to be of smaller size than Murrah and Nili-Ravi. *Nivsarkar, Vij & Tantia, 2000* in Nili-Ravi reported average HT, CG, and BL as 134.2, 207.7 and 165.4 cm, respectively, which is comparable to

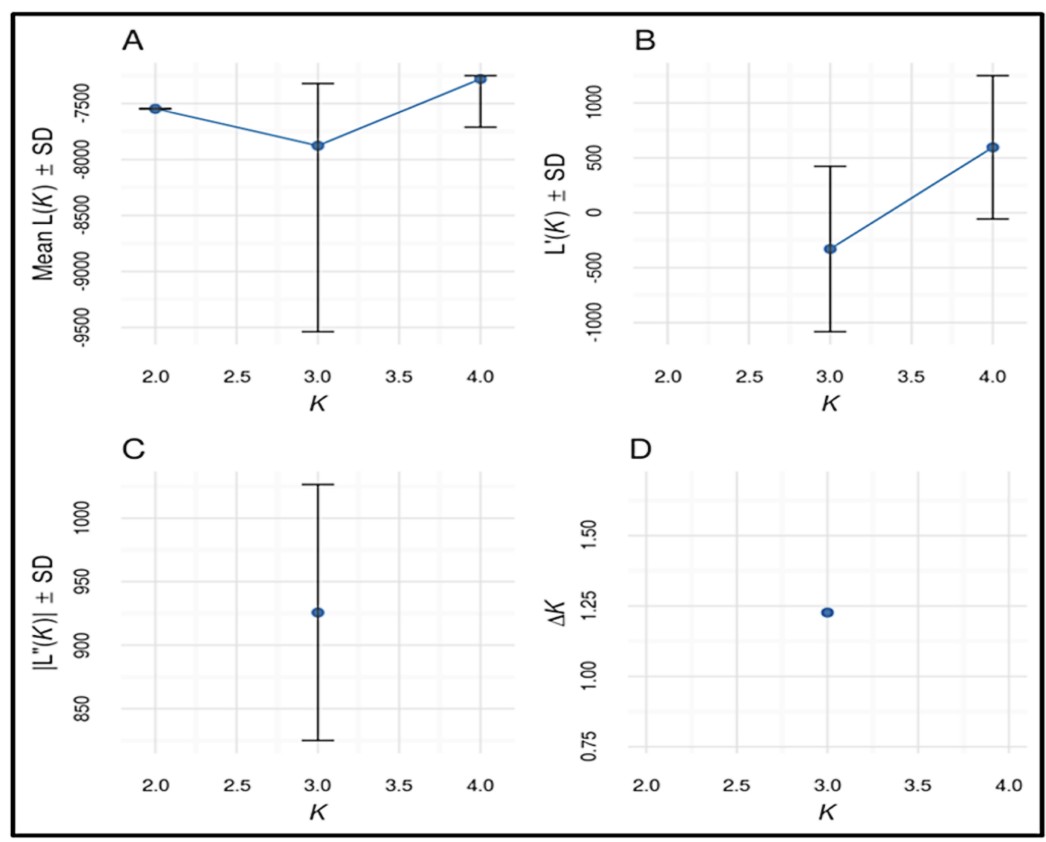

**Figure 4 Estimates of number of sub-populations (*K*) using different statistics by Evanno method to determine ideal number of clusters present in the studied buffalo populations.**

our results. CV% was highest for HL in Gojri (19.52%) and Murrah (12.61%) buffaloes indicating lesser selection pressure on them and more environmental influence. Face width (FW) was least variable in Murrah and Nili-Ravi while it varied greatly in Gojri buffaloes. Most of the body biometric traits measured were less variable indicating their reliability in population classification studies.

In the canonical discriminant analysis (refer to Table 2), two functions were needed for separation of three distinct population (*Asamoah-Boaheng & Sam, 2016*) and the first function (function 1) explains 66.7% of the variance and has a Wilk's lambda (0.008) with $p = 0.0001$. The second function explains only 33.3% of the variance in the data, with a recorded $p = 0.0001$ for Wilk's lambda (0.122). Wilks' Lambda value close to zero represents a greater number of variables contribute to the discriminant function (*Toalombo Vargas et al., 2019*), thus the first function in this study plays major role in classifying the breeds.

## Microsatellite variations and Genetic diversity

Microsatellite marker data being the best-suited molecular information for the assessment of genetic diversity (*Bowcock et al., 1994*; *Laval et al., 2000*; *Groeneveld et al., 2010*), allows future management and conservation of the breeds based on their genetic architecture

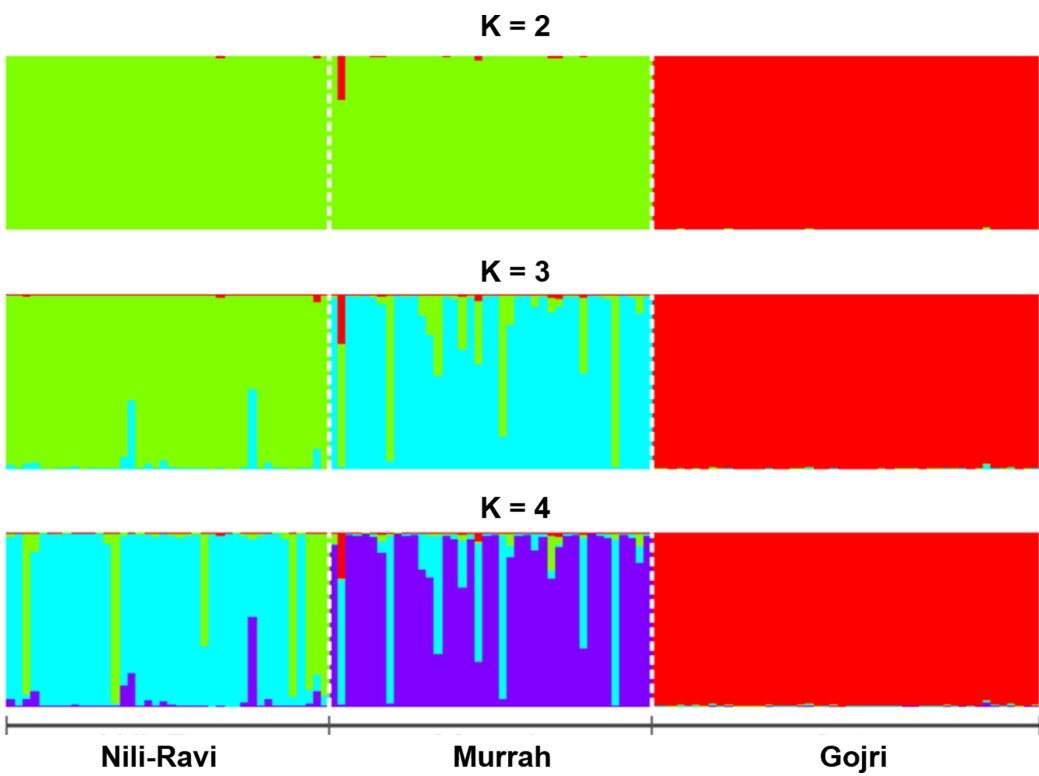

**K = 2**

**K = 3**

**K = 4**

Nili-Ravi          Murrah          Gojri

**Figure 5 Bayesian clustering of North-Indian buffalo populations under the assumption of $K = 2$–4 using STRUCTURE program reveals genetic admixture and introgression among Murrah and Nili-Ravi populations while Gojri buffalo is genetically distinct.** Each vertical bar represents individuals displaying membership coefficients for each population cluster. Populations are separated by dashed white lines. Graphics were obtained with CLUMPP (*Jakobsson & Rosenberg, 2007*).

(*Luikart et al., 2003*; *Taberlet et al., 2008*; *Toro, Fernández & Caballero, 2009*; *Teneva et al., 2013*). The FAO and the ISAG/FAO Advisory Group on Animal Genetic Diversity have proposed a panel of 25 SSR markers for diversity studies in buffaloes (*Singh et al., 2018*). Hence, in the present study the 22 highly polymorphic microsatellite markers out of 25 marker panel, were used for diversity analysis.

The mean number of alleles ($N_a$) in population over a range of loci is considered a fair indicator of allelic variation. The mean $N_a$ ranged from 0–10, 0–13 and 3–14 in Nili-Ravi, Murrah and Gojri buffaloes, respectively (refer to Table 3). The mean $N_a$ per locus for each population in the present study is similar to the reports of *Kathiravan et al. (2010)* in South Kanara buffaloes; *Marques et al. (2011)* in Brazilian buffaloes, *Martínez et al. (2009)*, *Bhuyan et al. (2010)* in Murrah buffaloes and in Purnathadi buffaloes *Ali et al. (2020)*. However, a higher number of alleles per locus ranged from 11–26 alleles in Indian water buffaloes have been reported by *Vijh et al. (2008)*. The type of breed under investigation, usage of the particular panel of microsatellite markers, methods of genotyping and the genetic polymorphism within the breed itself greatly influence this variation in the $N_a$.

For microsatellite data, *Ohta & Kimura (1973)* have established the relationship between the expected homozygosity and its estimator $\theta$, under a pure stepwise mutation

**Table 5 Individual wise ancestry level inferred through Bayesian method in Structure.**

| Label | Population | Inferred clusters | | |
|---|---|---|---|---|
| | | Goj | NR | MU |
| NR1 | 1 | 0.002 | 0.969 | 0.029 |
| NR2 | 1 | 0.002 | 0.988 | 0.011 |
| NR3 | 1 | 0.008 | 0.958 | 0.034 |
| NR4 | 1 | 0.004 | 0.958 | 0.038 |
| NR5 | 1 | 0.001 | 0.99 | 0.009 |
| NR6 | 1 | 0.002 | 0.994 | 0.004 |
| NR7 | 1 | 0.003 | 0.991 | 0.005 |
| NR8 | 1 | 0.002 | 0.992 | 0.006 |
| NR9 | 1 | 0.004 | 0.975 | 0.021 |
| NR10 | 1 | 0.002 | 0.994 | 0.004 |
| NR11 | 1 | 0.002 | 0.987 | 0.011 |
| NR12 | 1 | 0.002 | 0.992 | 0.006 |
| NR13 | 1 | 0.005 | 0.987 | 0.007 |
| NR14 | 1 | 0.004 | 0.991 | 0.005 |
| NR15 | 1 | 0.001 | 0.928 | 0.071 |
| NR16 | 1 | 0.002 | 0.601 | 0.397 |
| NR17 | 1 | 0.001 | 0.993 | 0.006 |
| NR18 | 1 | 0.004 | 0.959 | 0.037 |
| NR19 | 1 | 0.002 | 0.993 | 0.005 |
| NR20 | 1 | 0.002 | 0.947 | 0.051 |
| NR21 | 1 | 0.003 | 0.984 | 0.013 |
| NR22 | 1 | 0.005 | 0.99 | 0.005 |
| NR23 | 1 | 0.004 | 0.99 | 0.006 |
| NR24 | 1 | 0.002 | 0.993 | 0.005 |
| NR25 | 1 | 0.002 | 0.991 | 0.007 |
| NR26 | 1 | 0.004 | 0.993 | 0.003 |
| NR27 | 1 | 0.020 | 0.973 | 0.008 |
| NR28 | 1 | 0.002 | 0.996 | 0.003 |
| NR29 | 1 | 0.001 | 0.995 | 0.004 |
| NR30 | 1 | 0.003 | 0.981 | 0.015 |
| NR31 | 1 | 0.002 | 0.541 | 0.457 |
| NR32 | 1 | 0.002 | 0.993 | 0.005 |
| NR33 | 1 | 0.002 | 0.958 | 0.04 |
| NR34 | 1 | 0.002 | 0.993 | 0.006 |
| NR35 | 1 | 0.001 | 0.993 | 0.006 |
| NR36 | 1 | 0.003 | 0.994 | 0.004 |
| NR37 | 1 | 0.001 | 0.988 | 0.01 |
| NR38 | 1 | 0.003 | 0.986 | 0.011 |
| NR39 | 1 | 0.04 | 0.843 | 0.118 |
| NR40 | 1 | 0.003 | 0.994 | 0.003 |

(Continued)

| Table 5 (continued) | | | | |
|---|---|---|---|---|
| **Label** | **Population** | **Inferred clusters** | | |
| | | **Goj** | **NR** | **MU** |
| MU1 | 2 | 0.004 | 0.015 | 0.981 |
| MU2 | 2 | 0.28 | 0.707 | 0.013 |
| MU3 | 2 | 0.003 | 0.003 | 0.993 |
| MU4 | 2 | 0.005 | 0.007 | 0.988 |
| MU5 | 2 | 0.001 | 0.003 | 0.995 |
| MU6 | 2 | 0.005 | 0.013 | 0.983 |
| MU7 | 2 | 0.008 | 0.048 | 0.944 |
| MU8 | 2 | 0.004 | 0.946 | 0.05 |
| MU9 | 2 | 0.002 | 0.003 | 0.995 |
| MU10 | 2 | 0.003 | 0.01 | 0.987 |
| MU11 | 2 | 0.002 | 0.004 | 0.994 |
| MU12 | 2 | 0.002 | 0.119 | 0.879 |
| MU13 | 2 | 0.003 | 0.229 | 0.768 |
| MU14 | 2 | 0.003 | 0.461 | 0.536 |
| MU15 | 2 | 0.005 | 0.004 | 0.99 |
| MU16 | 2 | 0.005 | 0.019 | 0.976 |
| MU17 | 2 | 0.01 | 0.303 | 0.687 |
| MU18 | 2 | 0.002 | 0.018 | 0.979 |
| MU19 | 2 | 0.039 | 0.36 | 0.602 |
| MU20 | 2 | 0.002 | 0.006 | 0.992 |
| MU21 | 2 | 0.003 | 0.003 | 0.995 |
| MU22 | 2 | 0.004 | 0.804 | 0.192 |
| MU23 | 2 | 0.007 | 0.164 | 0.829 |
| MU24 | 2 | 0.003 | 0.007 | 0.99 |
| MU25 | 2 | 0.003 | 0.004 | 0.993 |
| MU26 | 2 | 0.002 | 0.065 | 0.933 |
| MU27 | 2 | 0.001 | 0.011 | 0.988 |
| MU28 | 2 | 0.017 | 0.084 | 0.899 |
| MU29 | 2 | 0.022 | 0.049 | 0.929 |
| MU30 | 2 | 0.002 | 0.011 | 0.987 |
| MU31 | 2 | 0.002 | 0.007 | 0.991 |
| MU32 | 2 | 0.014 | 0.438 | 0.548 |
| MU33 | 2 | 0.002 | 0.004 | 0.995 |
| MU34 | 2 | 0.002 | 0.009 | 0.99 |
| MU35 | 2 | 0.004 | 0.02 | 0.975 |
| MU36 | 2 | 0.004 | 0.983 | 0.013 |
| MU37 | 2 | 0.002 | 0.004 | 0.994 |
| MU38 | 2 | 0.005 | 0.015 | 0.98 |
| MU39 | 2 | 0.005 | 0.104 | 0.891 |
| MU40 | 2 | 0.002 | 0.01 | 0.988 |

| Table 5 (continued) | | | | |
|---|---|---|---|---|
| Label | Population | Inferred clusters | | |
| | | Goj | NR | MU |
| Goj1 | 3 | 0.995 | 0.002 | 0.003 |
| Goj2 | 3 | 0.995 | 0.002 | 0.003 |
| Goj3 | 3 | 0.996 | 0.002 | 0.001 |
| Goj4 | 3 | 0.991 | 0.006 | 0.003 |
| Goj5 | 3 | 0.996 | 0.002 | 0.002 |
| Goj6 | 3 | 0.993 | 0.004 | 0.002 |
| Goj7 | 3 | 0.997 | 0.001 | 0.002 |
| Goj8 | 3 | 0.988 | 0.008 | 0.004 |
| Goj9 | 3 | 0.995 | 0.002 | 0.003 |
| Goj10 | 3 | 0.989 | 0.006 | 0.005 |
| Goj11 | 3 | 0.997 | 0.001 | 0.002 |
| Goj12 | 3 | 0.997 | 0.001 | 0.002 |
| Goj13 | 3 | 0.997 | 0.001 | 0.002 |
| Goj14 | 3 | 0.997 | 0.001 | 0.002 |
| Goj15 | 3 | 0.991 | 0.003 | 0.006 |
| Goj16 | 3 | 0.995 | 0.002 | 0.002 |
| Goj17 | 3 | 0.996 | 0.002 | 0.002 |
| Goj18 | 3 | 0.995 | 0.003 | 0.002 |
| Goj19 | 3 | 0.991 | 0.005 | 0.004 |
| Goj20 | 3 | 0.986 | 0.009 | 0.005 |
| Goj21 | 3 | 0.997 | 0.001 | 0.002 |
| Goj22 | 3 | 0.996 | 0.002 | 0.002 |
| Goj23 | 3 | 0.99 | 0.004 | 0.006 |
| Goj24 | 3 | 0.997 | 0.001 | 0.001 |
| Goj25 | 3 | 0.996 | 0.002 | 0.002 |
| Goj26 | 3 | 0.996 | 0.002 | 0.002 |
| Goj27 | 3 | 0.997 | 0.002 | 0.002 |
| Goj28 | 3 | 0.995 | 0.002 | 0.002 |
| Goj29 | 3 | 0.995 | 0.002 | 0.003 |
| Goj30 | 3 | 0.995 | 0.002 | 0.002 |
| Goj31 | 3 | 0.995 | 0.002 | 0.003 |
| Goj32 | 3 | 0.994 | 0.003 | 0.004 |
| Goj33 | 3 | 0.992 | 0.004 | 0.004 |
| Goj34 | 3 | 0.996 | 0.002 | 0.002 |
| Goj35 | 3 | 0.995 | 0.002 | 0.003 |
| Goj36 | 3 | 0.995 | 0.002 | 0.003 |
| Goj37 | 3 | 0.992 | 0.004 | 0.005 |
| Goj38 | 3 | 0.995 | 0.002 | 0.003 |
| Goj39 | 3 | 0.992 | 0.001 | 0.006 |
| Goj40 | 3 | 0.996 | 0.002 | 0.002 |

| Label | Population | Inferred clusters | | |
|-------|-----------|-------------------|---|---|
| | | Goj | NR | MU |
| Goj41 | 3 | 0.994 | 0.003 | 0.004 |
| Goj42 | 3 | 0.967 | 0.006 | 0.026 |
| Goj43 | 3 | 0.989 | 0.003 | 0.008 |
| Goj44 | 3 | 0.994 | 0.003 | 0.003 |
| Goj45 | 3 | 0.988 | 0.006 | 0.006 |
| Goj46 | 3 | 0.997 | 0.001 | 0.002 |
| Goj47 | 3 | 0.993 | 0.003 | 0.004 |
| Goj48 | 3 | 0.991 | 0.002 | 0.007 |

Note:
NR, MU and Goj represent Nili-Ravi, Murrah and Gojri, respectively.

model *i.e.* expected homozygosity $= 1/\sqrt{1 + 2\theta}$. An estimator of $\theta$ can be obtained from microsatellite data by applying the formula, $\theta_{H} = [1/ (1 - H_{e})^{2} - 1]$ (*Excoffier & Lischer, 2010*), where $H_{e}$ is the expected heterozygosity. The mean $H_{e}$ ranging from 0.14 to 0.81 across all three population over all loci (refer to Table 3) is indicative of sufficient polymorphism to measure genetic variation (*Takezaki & Nei, 1996*). In Gojri buffaloes the expected heterozygosity ($H_{e}$) ranged from 0.12 to 0.82 that was comparable with the results reported by *Singh et al. (2019)*. While in both Murrah and Nili-Ravi, it ranged from 0 to 0.81 (refer to Table 3). Similar high overall mean $H_{e}$ were reported in Pandharpuri (*Khade et al., 2019*), Mehsana (*Jakhesara et al., 2010*), Egyptian (*Attia, Abou-Bakr & Hafez, 2014*) and Purnathadi (*Ali et al., 2020*) buffaloes. The substantially high $H_{e}$ values implies the presence of high genetic variability in the studied buffalo breeds and suitability of the marker panel for the present study.

The average F statistics over 19 loci were $F_{IS} = 0.11744$, $F_{ST} = 0.18252$ and $F_{IT} = 0.27852$ (refer to Table 4). In the present study, considerable degree of differentiation has been estimated compared to other buffalo populations from different regions, probably because these populations are genetically distinct. *Joshi et al. (2012)* reported an $F_{ST}$ value of 7.2% in buffaloes of Indo-gangetic plain, while *Vijh et al. (2008)* reported a value of 9.69%. However, a comparatively lesser value in eight Indian riverine buffalo was reported by *Kumar et al. (2006)* which was 3.4%. This value suggested the existence of greater genetic differentiation among North-Indian buffalo breeds than breeds found all over India. A heterozygote deficiency was evident from the positive mean $F_{IS}$ value (0.117 > 0) indicating low to moderate amount of inbreeding in the population. This could be attributed to assortative mating in small herds owned by farmers, genetic hitchhiking, or the null alleles (*Mishra et al., 2008*). However, AMOVA over all 22 loci showed 23.59% of variations between populations suggesting the distinctness of all three breeds. The $F_{IS}$ value was found to be 4.74%, which is comparable to values obtained in Purnathadi buffaloes (*Ali et al., 2020*).

The pair-wise $F_{ST}$ values ranged from 0.09 between Murrah and Nili-Ravi to 0.32 between Nili-Ravi and Gojri breeds. The $F_{ST}$ between Murrah and Gojri was 0.25 (Fig. 2A). Least differentiation was found between Nili-Ravi and Murrah (0.09) based on Slatkin linearized $F_{ST}$ while it was highest between Nili-Ravi and Gojri (0.46). Between Murrah and Gojri it was found to be 0.33 (Fig. 2B). Nei's distance (d); average within and between populations differentiation is presented in the form of a heat map (Fig. 2C), that shows least distance between Murrah and Nili-Ravi breeds and discriminate Gojri as another population. These results were also in compliance with the results from the factorial correspondence analysis based on molecular data and phylogenetic tree obtained from dissimilarity matrix. In the scatter plot of factorial analysis, Murrah and Nili-Ravi are invariably clustered together. Meanwhile, Gojri was found to be plotted on the opposite side of axis-2 (Fig. 3), yet with more scattering among individuals.

The linkage disequilibrium method relies on measures of departure from expected genotype and gametic frequencies, which is the basis for estimation of effective population size (*Hill, 1981*; *Waples, 1991*; *Luikart et al., 2010*). The $N_e$ estimated from microsatellite data reflects the true population distribution of North Indian buffaloes. The comparatively higher $N_e$ of Murrah buffaloes is due to the larger population distribution of the breed in India. The $N_e$ estimates of Gojri population reflects its present status and probable serious inbreeding in future. Hence, the ongoing indiscriminate breeding practices should be shifted to implementation of organized breeding policies focussed on conservation of this distinct breed.

**Bayesian genetic structure**

Structure software (*Pritchard, Stephens & Donnelly, 2000*) was used to determine the unbiased structure assuming no prior knowledge regarding the number of breeds. The highest delta $K$ ($\Delta K$) value was calculated as previously described (*Evanno, Regnaut & Goudet, 2005*). The optimum $\Delta K$ value (Fig. 4), was found at $K = 3$. For $K = 2$, there was no differentiation between Nili-Ravi and Murrah breed. One individual from Murrah population showed significant level of admixture from Gojri population. 99.7% of Gojri buffaloes were classified as a different breed whereas 99.7% and 98.9% of Nili-Ravi and Murrah buffaloes were assigned to one single population, respectively. When $K$ is assumed to be four, Gojri buffaloes are assigned to one distinct cluster with 99% of memberships. Structure assigned all three population into three different breeds when $K$ was assumed to be three (Fig. 5). This indicated that studied populations has well differentiated and possess unique allelic combinations despite being reared in similar geographical regions. However, a low to moderate amount of admixture could be observed in both Murrah and Nili-Ravi population. For $K = 3$, Nili-Ravi showed an average admixture of 3.7% from Murrah and 0.4% from Gojri buffaloes while it was quite high for Murrah with an average admixture of 15.2% and 1.2% from Nili-Ravi and Gojri, respectively. While Gojri population was found to have 99.4% pure blood with an admixture of 0.3% from Nili-Ravi and 0.4% from Murrah. Our results indicate the presence of sufficiently large genetic variability among the North Indian Riverine buffaloes. However, Gojri buffalo populations is unique, compared to Murrah and Nili Ravi buffalos, which were found to be

genetically closer than expected. Presently the breeding areas of all these populations are overlapping due to adoption of Murrah as an improver breed for milk production, thus leading to its dominance over Nili-Ravi and Gojri buffalo.

## CONCLUSION

This study demonstrated that the characterization and classification of genetic diversity in Indian buffaloes could be better accomplished through a parallel approach comprising morphometric traits and microsatellite markers. Study of buffalo genetic diversity of Northern India revealed admixture of two major dairy buffalo breeds and a distinct buffalo population was identified. The results obtained provides an opportunity for the design of genetic improvement programs with appropriate choice of breeds for upgrading local non-descript buffaloes along with conservation of unique germplasm. The estimates of effective population size and fixation indices indicate absence of intense systematic selection in past. Further studies involving large populations including samples from other regions of Indian buffalo with FAO recommended microsatellite loci are required to understand the genetic relationships among buffalo genetic resource of India.

## ACKNOWLEDGEMENTS

The authors wish to thank the Directors of ICAR-NDRI and ICAR-NBAGR for providing logistics, and infrastructural facilities, support and suggestions in completing this work. Technical support received from Mr. Subhash Chander, T-5, NBAGR, is gratefully appreciated.

### Funding

The authors received no funding for this work.

### Competing Interests

The authors declare that they have no competing interests.

### Author Contributions

- Vikas Vohra conceived and designed the experiments, analyzed the data, authored or reviewed drafts of the paper, and approved the final draft.
- Narendra Pratap Singh performed the experiments, analyzed the data, prepared figures and/or tables, and approved the final draft.
- Supriya Chhotaray analyzed the data, prepared figures and/or tables, and approved the final draft.
- Varinder Singh Raina performed the experiments, authored or reviewed drafts of the paper, and approved the final draft.
- Alka Chopra performed the experiments, prepared figures and/or tables, and approved the final draft.
- Ranjit Singh Kataria conceived and designed the experiments, performed the experiments, authored or reviewed drafts of the paper, and approved the final draft.

## Animal Ethics

The following information was supplied relating to ethical approvals (*i.e.*, approving body and any reference numbers):

ICAR-National Dairy Research Institute-IAEC (1705/GO/ac/13/CPCSEA) approved the experiment.

## Field Study Permissions

The following information was supplied relating to field study approvals (*i.e.*, approving body and any reference numbers):

For research purposes, a field permit is not required, however, only blood collection was done from the field herds with the consent of the herd owners/farmers under the supervision of a trained Veterinarian.

## Data Availability

The raw data is available in the Supplementary File.

## Supplemental Information

Supplemental information for this article can be found online at http://dx.doi.org/10.7717/peerj.11846#supplemental-information.

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
