# Peer review of "Morphometric and microsatellite-based comparative genetic diversity analysis in Bubalus bubalis from North India"

_PeerJ, doi:10.7717/peerj.11846_

## Round 0.1 · original submission · Major Revisions

Reviewers have provided very useful feedback to this important work, requesting revisions. Please, kindly attend to them, and provide the details required. Looking forward to your revised manuscript. Thank you very much.

Reviewer 1 ·

Basic reporting

Line 21: Please, add "the" before "prevailing".
Line 23: Replace "analysed" with "analyzed".
Line 25: Replace "size" with "sizes" or with "sizes for each breed".
Line 27: Replace "population" with "populations". Also, replace "Bayesian" with "The Bayesian".
Line 28: Add "the" before "Murrah". Also, replace "population" with "populations".
Line 32: Replace "Microsatellite Repeats" with "Microsatellite markers" or just "Microsatellite"
Line 40: Replace "includes" with "include".
Line 41: Remove "a" before "dominating".
Line 42: Replace "milk producing" with "milk-producing". Also, replace "the need of" with "the need for".
Line 45: Add "a" before "semi-migratory".
Line 46: Replace "requires" with "require".
Line 61: Remove "but" and add a comma (,) after "isolation".
Line 63: Replace "areas" with "area". Also, remove "as".
Line 66 and 71: Change to "well-recognized" and "region-specific".
Line 69: Remove "the" before "future".
Line 80: Figures should be self-informative and can be understood without referring to the main text. In Fig. 1, states are not mentioned, also colors for each breed are not clear enough to distinguish.
Line 85-86: Remove "average" and correct the sentence.
Line 95: Replace "To avoid age and sex effects" with "To avoid age effect".
Line 153: Measuring unit is not mentioned in Table 1.
Line 162: Put "Fig. 2" in parentheses. Please, do this for all other references to the pictures (e.g. lines 168, 172, 185, etc.)
Line 165: Remove "number of".
Line 168: How did you connect locus name in Fig. S1 with microsatellite names. It's not clear from the figure.
Line 173: Remove "by".
* Figures have many problems: Text in figures are not readable enough, even by zooming. Colors are not distinguishable enough. Almost all pictures have a lot of white space that should be cropped. Lacking adequate resolution.
* Tables are not uniform.

Experimental design

Experimental design sounds correct.
Line 86: How you investigated that the samples are unrelated?

Validity of the findings

No comment.

Additional comments

The manuscript needs a review and improvement on English grammar and writing.

·

Basic reporting

1.The major limitation of the study is the readability of the manuscript. The authors are advised to take English language editing services or consult the native English writers to edit the manuscript.

2.The authors are advised to follow and be consistent with the journal style for citations, bibliography, figures, tables, headings and subheadings.

3.The labels of figures must explain the figure well. For instance, in figure 6, explain each of the panel for K = 2 to 4. Put K = 2 to K= 4 on the top of the panel in vertical order. Explain well what the vertical line represents and how admixture patterns can be visualized.

4. Redraw Figure 3. It is hard to see the images. Also, check whether there is other elegant way of representing heat-map.


5. Report the descriptive statistics of the microsatellite loci in the main table.

Experimental design

1. Please mention the pedigree information of the sampled buffaloes. How many generation were those animals unrelated ?

2. Please mention- how the sampled buffaloes represent the entire breeds respectively? Given a very limited sample size collected from few regions- how can the study claim that the genetic differentiation of these sampled animals can represent the genetic differentiation of the entire breed ?

3. Please mention clearly the version control of all the softwares packages used for the analysis in this study.

Validity of the findings

no comment

Additional comments

Introduction

Line 21: remove prevailing

Line 22: I would suggest replacing distinctness with "distinct genetic entities"

Line 23-25: I would suggest changing the sentence to read: "Analysis of molecular variance revealed 81.8% of genetic variance was found within breeds, while 18.8% of the genetic variation was found among breeds that could differentiate the studied buffaloes breeds into 3 sub-populations".

Line 29; replace intermixing with admixture

Line 34: check the journal style for subheading

Line 35-37: Please edit the grammar and flow of the sentence

Line 39: I would suggest changing "significantly .... production" to "largest milk producing region in the country".

Line 40 replace "buffalo" with "buffaloes"

Line 43: replace "lesser-known" with "little-known"

Line 44: replace " which is having" by "with"

Line 48: remove "the"

Line 48: add "and" before genetic

Line 62-65: Please edit the grammar and flow of the sentence. Take help of native english writer or english language editing services, if needed.

Line 65: I would suggest changing the sentence to read as "The present study was performed to assess the levels of genetic diversity, and population structure among three buffalo breeds of North India".

Line 69: replace "can reduce the" with "reducing"

Materials and Methods

Line 72: check the journal style for subheading

Line 75: remove comma (,)

Line 75-76: replace "While the sampling for" with " , and samples for Nili-Ravi buffaloes were collected from Punjab state ( 28° 17' to 32° 17' N and 74° to 76° 41' E).

Line 77: remove "a" before smaller

Line 79: replace "thus" with ", and hence"

Line 82-83: Please edit the grammar and flow of the sentence. Take help of native english writer or english language editing services, if needed.

Line 84: replace "experimentation" with "studies"

Line88: Please mention- how the sampled buffaloes represent the breeds respectively?

Line 90. Please check the text citations.

Line 91-92: Please edit the grammar of the sentence. Take help of native english writer or english language editing services, if needed.

Line 96: Add comma after effects

Line 103: check the citation style of the journal

Line 107: replace 'sizing' with 'length'

Line 118: replace 'sizing' with 'length'

Line 121: Check the journal citation style

Line 121: replace 'sizing' with 'length'

Line 126: Check the journal citation style

Line 127: add "and" before allowing

Line 131: Remove comma before SPSS


Results

Line 151: check the journal style for subheading

Line 152: I would suggest changing the sentence to read as "The means and standard deviation, coefficient of variations and comparison of mean difference between population for each trait across population is listed ..."

Line 154-155: remove " of studied populations"

Line 156: Add % after 66.7

Line 162: remove comma and add parentheses for Fig. 2. . Check the journal style

Line 168: Check the journal style for inserting figure in the text

Line 171 to 172: Please edit the grammar of the sentence

Line 172: Check the journal style for inserting figure in the text

Line 172: Define θH before using the abbreviation. Do not start a sentence with an abbreviation.


Line 185-186: Check the journal style for inserting figure in the text

Line 188: replace clubbed with clustered; replace outgroup with "distinct group"

Line 188: Please edit the grammar and flow of the sentence

Line 193: check the grammar and flow of the sentence

Line 202: Report clustering of the bovine breeds obtained by STRUCTURE analyses in the main table

Discussion

Line 203: check the journal style for subheading

Line 204: remove available

Line 206: replace buffalo with buffaloes

Line 206: replace presently with currently

Line 208-210: Please edit the grammar of the sentence. Sentence construct is poor. Take help of native english writer or english language editing services, if needed.

Line 215: remove " extant ... till primarily"

Line 217: replace among with across

Line 220: Do not start a sentence with abbreviation

Line 221: replace varied with vary

Line 224: add of after average

Line 228: Do not start a sentence with abbreviation

Line 258: Check the journal format for inserting equation

Line 281: Check the journal style for inserting figure in the text

Line 289: replace clubbed with clustered

Line 297: Check the grammar and flow of the sentence

Line 298:check the journal style for subheading

Line 301: Check the journal style for inserting figure in the text

Line 304: remove comma and replace while with whereas

Line 308: remove of

Line 309: replace Still with "However,"


Conclusion

Line 319: check the journal style for subheading

Line 323: Please edit the grammar of the sentence

Line 324: replace creates with provides

Line 329: remove ", in a better perspective".

---

## Round 0.2 · Minor Revisions

Reviewers have considered accepting your work, given that they are satisfied with the revisions you have made. Before accepting this work, the editor would suggest the authors to kindly address the following:

a) Abstract needs to be revised, to capture clearly the rationale for the study, objective statement (consistent with introduction), the key aspect of method, essential results (which should connect with the conclusions).

b) Introduction: Kindly delete the very first sentence, as well as all information that has mentioned milk, because it is not relevant to this study. The introduction should be revised as follows, in four paragraphs :
- The importance of genetic diversity in a given animal population, what brings about genetic diversity, how does it evolve over time, which animals have such been studied (there are a lot, so just mention a few and narrow it down to buffaloes)
- Now, talk about buffaloes in the world, and narrow it down to that of India, which aspects of `India do these species avail? Genetic studies associated with buffaloes, and those that have been studied, then extract information from lines 38/39 to 48 to make up this paragraph
- The last two paragraphs are ok. However, the last (or last but one) paragraph should include information of previous morphometric and microsatellite-based comparative studies, and the kind of information it could help provide., including which statistical tool that has been used in such studies.
The editor will be looking out for these in its detail, so kindly make effort to provide sufficient information as requested.

c) In your statistical analysis, canonical discriminant analysis, Bayesian Markov chain, Monte Carlo approach, and principal coordinate analysis were employed. Please, kindly provide details after each sentence where it is first mentioned, what it is, why it is used, and how it is used. This is for readers who are not familiar with them to learn more about it. The editor will be looking out for this detail as well, kindly make effort to provide them.

d) In the results, all the places where statistical difference is shown, kindly indicate the exact p-value (not p<0.05), and include R-sq (adj) value, and F-change value (if available).

e) In the discussion, it is important to indicate (Refer to Table ??) or (Refer to Figure ??) at all the places where results of a specific Table or Figure are being discussed. Kindly make sure this is included, the editor will be looking out for this as well.

Look forward to seeing your revised manuscript. This is a very useful study. Thank you.

---

## Round 0.3 · accepted · Accept

The editor considers the revised manuscript very improved, as the authors have thoroughly and sufficiently addressed all concerns raised by the reviewers. The revised manuscript can now be accepted for publication. PeerJ appreciates the scholarly work of the authors. Thank you for finding PeerJ as your journal of choice. Looking forward to your future scholarly contributions. Congratulations and very best wishes :)